# MicroRNAs Associated with Chronic Mucus Hypersecretion in COPD Are Involved in Fibroblast–Epithelium Crosstalk

**DOI:** 10.3390/cells11030526

**Published:** 2022-02-02

**Authors:** Hataitip Tasena, Wim Timens, Maarten van den Berge, Joy van Broekhuizen, Brian K. Kennedy, Machteld N. Hylkema, Corry-Anke Brandsma, Irene H. Heijink

**Affiliations:** 1Department of Pathology and Medical Biology, University of Groningen, University Medical Centre Groningen, 9713 GZ Groningen, The Netherlands; h.tasena@nus.edu.sg (H.T.); w.timens@umcg.nl (W.T.); joyvanbroekhuizen@gmail.com (J.v.B.); m.n.hylkema@umcg.nl (M.N.H.); c.a.brandsma@umcg.nl (C.-A.B.); 2Groningen Research Institute for Asthma and COPD, University of Groningen, University Medical Centre Groningen, 9713 GZ Groningen, The Netherlands; m.van.den.berge@umcg.nl; 3Healthy Longevity Translational Research Program, National University of Singapore, Singapore 117596, Singapore; bkennedy@nus.edu.sg; 4Department of Biochemistry, Yong Loo Lin School of Medicine, National University of Singapore, Singapore 117596, Singapore; 5Department of Pulmonary Diseases, University of Groningen, University Medical Centre Groningen, 9713 GZ Groningen, The Netherlands; 6Agency for Science, Technology and Research (A*STAR), Singapore Institute for Clinical Sciences, Singapore 117596, Singapore; 7Centre for Healthy Ageing, National University Health System, National University of Singapore, Singapore 117596, Singapore

**Keywords:** microRNA, chronic mucus hypersecretion, chronic obstructive pulmonary disease

## Abstract

We recently identified microRNAs (miRNAs) associated with chronic mucus hypersecretion (CMH) in chronic obstructive pulmonary disease (COPD), which were expressed in both airway epithelial cells and fibroblasts. We hypothesized that these miRNAs are involved in communication between fibroblasts and epithelium, contributing to airway remodeling and CMH in COPD. Primary bronchial epithelial cells (PBECs) differentiated at the air–liquid interface, and airway fibroblasts (PAFs) from severe COPD patients with CMH were cultured alone or together. RNA was isolated and miRNA expression assessed. miRNAs differentially expressed after co-culturing were studied functionally using overexpression with mimics in mucus-expressing human lung A549 epithelial cells or normal human lung fibroblasts. In PBECs, we observed higher miR-708-5pexpression upon co-culture with fibroblasts, and miR-708-5p expression decreased upon mucociliary differentiation. In PAFs, let-7a-5p, miR-31-5p and miR-146a-5p expression was significantly increased upon co-culture. miR-708-5p overexpression suppressed mucin 5AC (MUC5AC) secretion in A549, while let-7a-5poverexpression suppressed its target gene *COL4A1* in lung fibroblasts. Our findings suggest that let-7a-5p, miR-31-5p and miR-146a-5p may be involved in CMH via fibroblasts–epithelium crosstalk, including extracellular matrix gene regulation, while airway epithelial expression of miR-708-5p may be involved directly, regulating mucin production. These findings shed light on miRNA-mediated mechanisms underlying CMH, an important symptom in COPD.

## 1. Introduction

One of the characteristics of chronic obstructive pulmonary disease (COPD) is chronic mucus hypersecretion (CMH). CMH is characterized by the presence of chronic inflammation, chronic cough and exaggerated sputum production [1] and is associated with lower quality of life and higher mortality. MicroRNAs (miRNAs) regulate many cellular processes in health and disease, and our previous findings suggest their involvement in the pathogenesis of CMH [2]. MiRNAs are small non-coding RNA molecules consisting of approximately 22 nucleotides, which post-transcriptionally regulate gene expression by inducing mRNA degradation or inhibiting protein translation [3]. Over 60% of mammalian genes are predicted to be targeted by miRNAs [3]. They have been reported to be involved in various respiratory diseases, including COPD [4]. Previously, we identified 10 miRNAs that are associated with CMH in COPD using miRNA gene expression profiles of bronchial biopsies [2]. Among these miRNAs, the expression of let-7a-5p, let-7d-5p, let-7f-5p, miR-31-5p and miR-708-5p was higher with CMH, and the expression of miR-134-5p, miR-146a-5p and miR-193-5p, miR-500a-3p and miR-1207-5p was lower with CMH.

It is not yet clear whether these miRNAs are involved in dysfunctional fibroblast–epithelial cross-talk, contributing to CMH. Previously, it has been suggested that stromal cells, such as airway smooth muscle cells (ASMCs) and fibroblasts, play a crucial role in epithelial homeostasis and differentiation [5,6,7,8]. More recently, our group studied dysregulated crosstalk between epithelial cells and fibroblasts in COPD in a co-culture model, and we identified regulatory roles for epithelial-derived IL-1α and fibroblast-expressed miR-146a-5p [9], one of the CMH-associated miRNAs. We showed changes in extracellular matrix gene expression in fibroblasts, as well as pro-inflammatory mediators upon co-culture with epithelial cells, which were driven by IL-1α, indicating that epithelial cells can stimulate extracellular matrix (ECM) changes in the underlying mesenchymal cells. Furthermore, our group previously showed that fibroblast mediators increase mucus production by epithelial cells [10]. Altogether, these findings suggest that fibroblast–epithelial cell crosstalk may contribute to changes in the airway wall and airway epithelium involved in CMH development in COPD. We hypothesized that the CMH-associated miRNAs that we previously identified in bronchial biopsies [2] are involved in aberrant fibroblast–epithelial cell crosstalk in CMH. Using a co-culture model in which mucociliary-differentiated primary bronchial epithelial cells (PBECs) were co-cultured with primary airway fibroblasts (PAFs) for 1 week during air–liquid-interface- (ALI) differentiation, this study aimed to investigate which and how CMH-associated miRNAs are involved in fibroblast–epithelium crosstalk, eventually contributing to mucus hypersecretion. We functionally assessed the effects of differentially expressed miRNAs in co-cultured PBECs and PAFs vs. mono-cultured cells using overexpression with mimics in mucus-producing human lung A549 epithelial cells and human lung fibroblasts.

## 2. Materials and Methods

### 2.1. Mono-Culture and Co-Culture of Air–Liquid-Interface- (ALI) Differentiated PBECs and PAFs

PBECs were isolated from explanted lungs of 3 COPD stage IV patients with CMH undergoing lung transplantation, as previously described [11]. In addition, PBECs were isolated from residual bronchial tissue of 6 non-COPD transplant donors, of whom no further information was available [12]. PAFs were isolated from 6 COPD stage IV patients with CMH as previously published [13]. Patient characteristics are described in Table 1. COPD-derived PBECs and PAFs used in this study are derived from left-over lung material after lung surgery and transplant procedures. This material was de-identified and not subject to the act on medical research involving human subjects in The Netherlands, and therefore no specific ethical approval and informed consent was obtained. This study was conducted according the national ethical and professional guidelines (https://www.federa.org accessed on 23 November 2021), the national law, the General Data Protection Regulation (EU) 2016/679 (GDPR) and the Research Code of the University Medical Center Groningen (https://umcgresearch.org/en-GB/w/research-code-umcg accessed on 23 November 2021).

The presence of CMH was based on clinical records, with patients indicating daily cough and production of mucus. In this study, the information on CMH was collected retrospectively from the clinical records of the patients, which were checked for information regarding complaints and symptoms on daily cough and sputum production. We only selected patients of whom information on CMH symptoms was available. PBECs were grown in Keratinocyte Serum-Free Growth Medium (KSFM, Gibco, Grand Island, NY, USA) supplemented with 1% Penicillin (10,000 U/mL)/Streptomycin (10,000 μg/mL) (P/S) (Gibco, Grand Island, NY, USA), 1 μM isoproterenol, 25 ng/mL bovine pituitary extract (BPE, Gibco, Grand Island, NY, USA) and 2.5 µg/mL epidermal growth factor (EGF, Gibco, Grand Island, NY, USA). PAFs were grown in HAM’s F12 medium supplemented with 1% penicillin/streptomycin, 1% L-glutamine (Lonza, Basel, Switzerland) and 10% fetal bovine serum (FBS, Sigma-Aldrich, Darmstadt, Germany). Cells were passaged when 90% confluent. In all experiments, we used PBECs in passage 3 and PAFs in passages 4–6. The patient characteristics are described in Table 1.

ALI culture of PBECs was performed as previously described [10], and successful differentiation was confirmed by the expression of mucus and substantial increase in trans-epithelial resistance. Transwell inserts for 24-well plates (Corning^®^, Corning, NY, USA) were coated with 10 μg/mL bovine serum albumin (BSA; Sigma-Aldrich, MO, USA), 10 μg/mL fibronectin (Sigma Aldrich, St. Louis, MO, USA) and 30 μg/mL collagen (PureCol^®^, Advanced Biomatrix, San Diego, CA, USA) in Eagle’s Minimum Essential Medium (EMEM, Lonza, Walkersville, MD, USA) before being used to culture PBECs. PBECs from each donor were seeded at a density of 75,000 cells/insert in ALI culture medium prepared by mixing DMEM (LONZA BE12-709F, Basel, Switzerland) and BEBM (Clonetics CC-3171, Lonza, Basel, Switzerland) in 1:1 ratio supplemented with a set of BEGM Single Quots (Clonetics CC-4175, Lonza, Basel, Switzerland) and 1.5 µg/mL BSA (Sigma-Aldrich, Darmstadt, Germany) and 15 ng/mL retinoic acid (Sigma-Aldrich, Darmstadt, Germany). After 4–5 days, the cells were air-exposed for 14 days (day 0–14), during which the basal medium was refreshed every 2–3 days. PAFs were seeded in 24-well plates in HAM’s F12 medium and cultured until 70–80% confluence. Co-cultures were established by placing the transwell inserts with PBECs onto the wells with PAFs, and co-culturing the cells from day 7 to day 14 using ALI culture medium. At the end of day 14, cells were harvested separately and the total RNA from PBECs and PAFs isolated. The experimental design of the co-cultures is demonstrated in Figure 1. Non-COPD control PBECs were cultured in ALI for 28 days, in the absence of PAFs but in the presence of interleukin (IL)-13 (1 ng/mL; Peprotech, Rocky Hill, NJ, USA) to enhance goblet cell differentiation, and RNAs were harvested on days 0, 14, 21 and 28 [12].

### 2.2. qPCR for miRNA Expression

To assess miRNA expression, total RNA was converted to cDNA using the TaqMan microRNA reverse transcription kit (Life Technologies, Bleiswijk, The Netherlands) and microRNA assays (Life Technologies, Bleiswijk, The Netherlands) for let-7a-5p (assay id: 000377), miR-31-5p (002279), miR-134-5p (000459), miR-146a-5p (000468), miR-193a-5p (002281), and miR-708-5p (002341). qPCR for miRNAs was performed using LightCycler^®^ 480 Probes Master according to the manufacturer’s guidelines (Roche, Basel, Switzerland). Expression of all miRNAs was normalized to the expression of small nuclear RNA, RNU48 (001006) [2,9].

### 2.3. MicroRNA Transfection of A549 Alveolar Epithelial Cells and Normal Human Lung Fibroblasts (NHLF)

The human lung adenocarcinoma cell line A549 was cultured in Roswell Park Memorial Institute (RPMI) 1640 (Lonza, Basel, Switzerland) medium, supplemented with growth medium with 10% fetal calf serum (FCS) (Sigma-Aldrich, Darmstadt, Germany) and 1% Penicillin (10,000 U/mL)/Streptomycin (10,000 μg/mL) (P/S) (Gibco, California, CA, USA). The cells were seeded at a density of 50,000 cells per cm2 in a 24-well plate in 4 independent experiments. The growth medium was replaced by quiescent medium (FCS absent) at 80% cell confluence. Transfection with 1 nM miR-708 mimic (Qiagen, Hilden, Germany) and 1 nM AllStars negative control siRNA (Qiagen, Hilden, Germany) was performed using Lipofectamine RNAiMAX (Invitrogen, California, CA, USA) according to the manufacturers protocol. Twenty-four hours after transfection, the FCS-free medium was refreshed. Twenty-four hours later, cell-free supernatant samples were collected and stored at –20 °C. Total RNA was isolated from the cells using TRI reagent (Molecular Research Center, Cincinnati, OH, USA) and stored at –80 °C.

Normal Human Lung Fibroblasts (NHLF, Lonza CC-2512, Basel, Switzerland), kindly donated by Dr. Daniel B.L. Teh, were cultured in Fibroblast Basal Medium (FBM) supplemented with FGMTM-2 SingleQuotsTM (Lonza, Basel, Switzerland). The cells were seeded in duplicates at a density of 40,000 cells per cm2 in a 24-well plate in the growth medium, and transfected with 5 nM let-7a-5p, miR-31-5p, or AllStars negative control siRNA using HiPerFect (Qiagen, Hilden, Germany) according to the manufacturer’s Fast Forward Protocol. Twelve hours after transfection, the medium was then refreshed by FBM supplemented with FGMTM-2 SingleQuotsTM except FCS. RNA samples were collected 24 h later using Purelink rna mini kit (Invitrogen, California, CA, USA) according to the manufacturer’s protocol. We performed 4 independent experiments with NHLF from the same donor.

### 2.4. qPCR to Validate miRNA Targets

To assess mRNA expression in A549 cells transfected with miRNA mimic, cDNA was synthesized with the iScript cDNA synthesis kit (BioRad, Hercules, CA, USA) in accordance with the manufacturer’s instructions. qPCR was performed on technical duplicates using TaqMan (Life Technologies, Waltham, MA, USA) in accordance with the manufacturers’ respective instructions, using validated primers to assess *Forkhead box protein A2* (*FOXA2), transmembrane protein 88 (TMEM88), β2-microglobulin (B2M**) and*
*Peptidyl**-prolyl cis-trans isomerase A* (*PPIA*) expression. To assess mRNA expression in NHLFs transfected with miRNA mimics, a High-Capacity cDNA Reverse Transcription Kit (Applied Biosystems, Foster City, CA, USA) was used, followed by SYBR Green PCR according to the manufacturer’s guidelines. Sequences of qPCR primers (IDT, Singapore) are listed in Table 2.

### 2.5. Human MUC5AC Enzyme-Linked Immunosorbent Assay

Mucin 5AC (MUC5AC) levels were determined in supernatants using C96 Maxisorp NUNC Immuno-plate (Sigma-Aldrich, Darmstadt, Germany) coated with 100 µL/well of 500 ng/mL MUC5AC antibody (Thermo Fisher Scientific, Waltham, MA, USA) diluted in PBS (Gibco, California, CA, USA) overnight at room temperature on a platform shaker. After overnight incubation, the plate was washed three times with 0.05% Tween 20 (Sigma-Aldrich, Darmstadt, Germany) in PBS (Gibco, California, CA, USA) (wash buffer). Unbinding sites were blocked with 300 μL/well of 2% BSA (Sigma-Aldrich) and 0.05% Tween 20 (Sigma-Aldrich, Darmstadt, Germany) in PBS (Gibco, California, CA, USA) (block buffer) for 2 h at room temperature on a platform shaker. After blocking, the plate was washed as previously and 100 μL/well MUC5AC, standard or sample, was added and incubated for 1 h at room temperature on a platform shaker. MUC5AC standard was prepared by making a 7× serial dilution of 24 h 5 ng/mL PMA- (Sigma-Aldrich, Darmstadt, Germany) stimulated A549 supernatant in wash buffer. Samples were diluted 1:5 in wash buffer. After sample incubation, the plate was washed as previously and incubated with 100 μL/well conjugate for 1 h at room temperature on a platform shaker. Conjugate was prepared by diluting 1 mg/mL SBA-HRP in wash buffer to a final concentration of 0.6 μg/mL. After conjugate inhibition, the plate was washed as previously, and the plate was incubated with 100 μL/well TMB substrate (BioRad, Hercules, CA, USA) at room temperature while being protected from direct light. The reaction was stopped by adding 100 μL/well 1.8M sulfuric acid, and the plate was read in the EL808 Ultra Microplate Reader (BioTek Instruments Inc., Winooski, VT, USA) at 450 nm.

To prepare the MUC5AC standard, A549 cells were grown until confluent in a T25 flask (Sigma-Aldrich, Darmstadt, Germany). The A549 cells were stimulated with 5 ng/mL PMA (Sigma-Aldrich, Darmstadt, Germany) for 24 h. After 24 h, supernatant was collected and stored at –20 °C. The MUC5AC standard was prepared by making a serial dilution of the supernatant, starting with undiluted supernatant as the first sample. As the concentration of MUC5AC in the samples of the standard is not known, MUC5AC concentration was set as 100% for the first sample in the standard, 50% for the second sample in the standard, 25% for the third sample in the standard, and so on. These values were used to determine the concentration of MUC5AC in the samples (arbitrary unit).

### 2.6. Statistical Analyses

A one-sample Wilcoxon signed rank test was used to determine significant differences between mono-cultured and co-cultured PBECs (*n* = 1) and PAFs (*n* = 6). A paired-samples Wilcoxon signed rank test was used for comparisons between mono-cultured and co-cultured PBECs (*n* = 3) and PAFs (*n* = 6). A paired Student’s t-test was used for comparisons between mimic and control oligos in the A549 cell line and NHLF. All statistical analyses were performed using GraphPad PRISM 9.

## 3. Results

### 3.1. Several CMH-Associated miRNAs Are Differentially Expressed upon Co-Culture

To determine which of the 10 CMH-associated miRNAs previously identified in bronchial biopsies [2] (Table 3) are likely involved in fibroblast-to-epithelial cell crosstalk, we first assessed whether the expression of any of these miRNAs would change upon co-culture. We co-cultured ALI-differentiated PBECs from one donor with PAFs from six different donors with COPD and CMH. Since let-7a-5p, let-7d-5p and let-7f-5p are from the same miRNA cluster, of which seed sequences are very similar, sharing several potential targets [2], let-7a-5p was selected as a representative of the let-7 family. Further, since we previously observed that miR-500a-3p and miR-1207-5p were neither expressed in PBECs nor PAFs [2], we did not include them in the current study.

We observed that all miRNAs except miR-134-5p were expressed by PBECs. While no change was observed for expression of let-7a-5p, miR-31-5p, miR-146a-5p and miR-193a-5p, the expression of miR-708-5p tended to increase upon co-culture with PAFs (Figure 2a). In PAFs, let-7a-5p, miR-31-5p, miR-708-5p, miR-134-5p, miR-146a-5p and miR-193a-5p were expressed in mono-culture as well as in co-culture, with a significantly higher expression of let-7a-5p and miR-146a-5p upon co-culture and a trend for higher expression of miR-31-5p (Figure 2b). We then selected the most differently expressed miRNAs for further validation using two additional PBEC donors co-cultured with the same six PAF donors. When combining the results from all three PBEC donors, we observed a significant increase in miR-708-5p expression upon co-culture with PAFs (Figure 2c). Further, the expression of let-7a-5p, miR-31-5p and miR-146a-5p in PAFs was significantly increased upon co-culture (Figure 2d).

### 3.2. Decrease in miR-708 Expression during Mucociliary Differentiation

Because of its positive association with CMH and its upregulation in epithelial cells upon co-culture, we hypothesized that miR-708-5p may be involved in the regulation of the differentiation towards mucus-producing cells and/or mucus production. Therefore, we next assessed miR-708-5p expression longitudinally during PBEC culture in ALI from day 0 to day 28 in PBECs from six non-COPD donors. To promote differentiation towards goblet cells, IL-13 was added, leading to a strong increase in MUC5AC expression and mucus production, as well as a decrease in MUC5AC suppressor FOXA2, as described before [12]. We observed a significant decrease in miR-708-5p during mucociliary differentiation, starting from day 14 (Figure 3), preceding the differentiation into mucus-producing cells [12].

### 3.3. Transfection with miR-708-5p Mimic Suppresses MUC5AC Secretion by Lung Epithelial Cells

To investigate whether the observed upregulation of miR-708 upon co-culture regulates mucus production, we aimed to overexpress miR-708-5p in ALI-differentiated airway primary epithelial cells. However, we were unable to induce a robust overexpression in these cells upon transfection with the mimic (Appendix A), and continued with the mucus-expressing cell line A549 as a model. Upon transient transfection of miR-708-5p mimic in A549, we assessed the effect on the validated target genes FOXA2 and TMEM88. FOXA2 expression was not significantly altered (Figure 4a), while TMEM88 expression was not detected. In addition, we assessed the secretion of MUC5AC, and observed that MUC5AC protein levels significantly reduced upon miR-708 overexpression in A549 cells (Figure 4b).

### 3.4. Let-7a-5p Overexpression Regulates COL4A1 Expression While Transfection with miR-31-5p Mimic Does Not Alter COL5A1 Expression in Lung Fibroblasts

Next, we assessed functional effects of the miRNAs differentially expressed in the fibroblasts upon co-culture. As we previously demonstrated that miR-146a-5p is involved in mesenchymal-epithelial crosstalk, regulating epithelial MUC5AC release by suppressing mesenchymal CCL20 release [8], we decided to focus on let-7a-5p and miR-31-5p in fibroblasts. We transfected NHLF cells with let-7a-5p and miR-31-5p mimic and assessed the effects on known predicted and correlated target genes that were also negatively associated with CMH as demonstrated previously [2]. For Let-7a-5p, we selected collagen type 4 alpha 1chain (COL4A1), COL4A2 and integrin subunit alpha 7 (ITGA7) and for miR-31-5p we selected COL5A1. After 24 h of transfection with the let-7a-5p mimic, expression of COL4A1 was significantly downregulated. A similar effect was observed for COL4A2 and ITGA7, with three out of four replicates showing decreased expression (Figure 5a), although this was not significant. The effects of the let-7a-5p mimic seemed to be specific for the selected target genes, as no effect of this mimic was observed on the expression of COL5A1 or COL1A1, the latter of which was added as a general ECM marker gene (Appendix A). Treatment with the miR-31-5p mimic did not significantly affect expression of COL5A1 (Figure 5b).

## 4. Discussion

In this study, we investigated the role of CMH-associated miRNAs in epithelial-fibroblast crosstalk and showed that in COPD-derived airway epithelial cells, miR-708-5p expression increased upon co-culture with COPD-derived airway fibroblasts, while no differences were observed for let-7a-5p, miR-31-5p, miR-146a-5p, miR-193a-5p and miR-134. In fibroblasts from these COPD patients with CMH, let-7a-5p, miR-31-5p and miR-146a-5p expression was significantly increased upon co-culture with airway epithelial cells. The increased expression of these CMH-associated miRNAs upon co-culture suggests their involvement in fibroblast–epithelial cell crosstalk, which may be dysregulated in COPD patients with CMH. Our findings support a regulatory role for miR-708-5p in mucus hypersecretion, with downregulated miR-708-5p expression during mucociliary differentiation, and decreased MUC5AC secretion upon miR-708 overexpression. Furthermore, our findings support a potential role for let-7a-5p in the regulation of basement membrane changes in CMH by regulating the expression of *COL4A1*.

In line with our findings, the expression of miR-708-5p was previously shown to be negatively associated with mucociliary differentiation at ALI [18], suggesting that it may negatively regulate mucociliary differentiation. Its downregulation may thus allow for differentiation towards goblet cells and/or production of mucus. To the best of our knowledge, our study is the first to demonstrate that miR-708 regulates epithelial mucus production. As miR-708 was positively associated with CMH [2], we speculate that miR-708 upregulation in CMH may act as a compensatory mechanism in the interaction between epithelial cells and fibroblasts as an attempt to suppress mucus hypersecretion. Whether and why this mechanism fails in CMH will need further investigation.

Furthermore, little is known about the role of let-7a-5p in relation to CMH development and fibroblast-epithelial cell crosstalk. Since let-7a-5p was only increased in fibroblasts upon co-culture of epithelial cells and fibroblasts, it is more likely involved in fibroblast-to-epithelial communication rather than directly regulating epithelial cell differentiation and/or mucus secretion. COL4A1 and COL4A2 are essential components of basement membrane [19] and are predicted targets of the let-7 family. We observed that their expression levels were negatively associated with let-7a-5p in COPD bronchial biopsies in our previous study [2]. In addition, *COL4A1* expression negatively correlated with the presence of CMH in COPD.

Thickness of the basement membrane is associated with an increase in submucosal glands and central airway remodeling in asthma [20], but this has not been linked with CMH in COPD. Here we show that overexpression of let-7a-5p in human lung fibroblasts resulted in significant downregulation of *COL4A1* and a similar effect for *COL4A2*. This is in accordance with our previous observations in bronchial biopsies of COPD patients [2], and indicates that higher let-7a expression in CMH may lead to lower *COL4A1* expression. Furthermore, Kumar et al. showed that intranasal administration of let-7 miRNA mimic suppresses IL-13 expression and attenuates mucus production in bronchial biopsies, indicating also a direct or indirect effect of let-7 on epithelial mucus production [21]. We speculate that the changes in let-7a-5p expression upon dysregulated fibroblast-epithelial communication may contribute to abnormalities in the basement membrane that may in turn affect epithelial differentiation and promote CMH.

Additionally, miR-31-5p was upregulated in co-cultured fibroblasts. This miRNA is of relevance to CMH, as miR-31-5p was also positively associated with CMH in asthma, suggesting that it may be a component of a shared mechanism between CMH in asthma and COPD [22]. One of the predicted targets of miR-31-5p that was also negatively associated with CMH in our previous study is *COL5A1*. Similarly to *COL4A, COL5A1* is expressed in the basement membrane and could thus affect epithelial differentiation. In the current experiments, however, no regulatory effect of miR-31-5p on *COL5A1* expression in human lung fibroblasts could be demonstrated.

In contrast to let-7a-5p and miR-31-p, miR-146a-5p expression was lower with CMH [2] and may function as negative feedback to suppress pro-inflammatory responses, i.e., secretion of IL-6, CXCL8 and CCL20 [9]. Both IL-6 and CXCL8 were shown to promote mucin secretion by airway epithelial cells upon differentiation in ALI culture [10]. Additionally, we previously showed that miR-146a-5p regulates CCL20 release upon mesenchymal–epithelial crosstalk and subsequent MUC5AC production [23].

In this study we did not address the mechanisms involved in the altered expression of miRNAs upon fibroblast-epithelial cell culture. Our previous study showed that epithelial release of IL-1α was responsible for the upregulation of miR-146a-5p in fibroblasts [9]. Whether a similar mechanism may be involved in the dysregulation of other CMH-associated miRNAs that were differently expressed upon co-culture needs further investigation. Transfer of miRNAs via extracellular vesicles constitutes an additional mechanism of intercellular communication, which is also of interest to further study.

Our study has some limitations. The effects of miRNA mimics were assessed in lung fibroblasts from one donor, or epithelial cells from an alveolar mucus-producing carcinoma cell line. In the future, it will be relevant to assess effects in airway-derived fibroblasts and use primary airway epithelial cells cultured at the air–liquid interface to study effects on epithelial mucus production and/or differentiation, although it is challenging to transfect these cells with mimics. Furthermore, it will be relevant to compare cells from donors with and without CMH. Finally, we only focused on one of the let-7 family members, and it will be of interest to study the functional role of the other members as well.

Overall, this study provides candidate miRNAs that may reflect and contribute to dysregulated fibroblast–epithelial cell crosstalk involved in CMH development in COPD. Additional insight in the mechanistic role of these miRNAs in CMH pathogenesis may reveal novel therapeutic options to target this important symptom of COPD.

## Figures and Tables

**Figure 1 cells-11-00526-f001:**
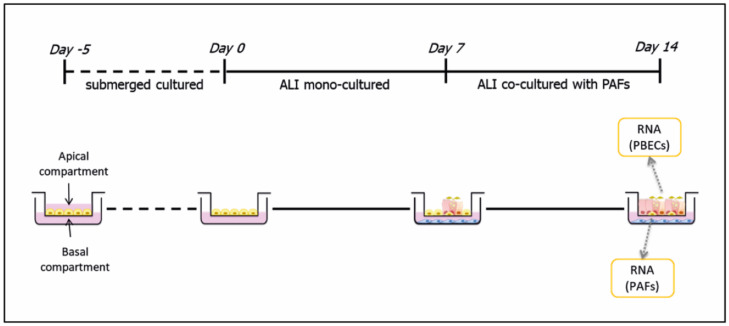
Co-culture of primary bronchial epithelial cells (PBECs) and primary airway fibroblasts (PAFs) in air–liquid interface (ALI). PBECs were submerged-cultured for 5 days before being air-exposed for 2 weeks (days 0–14), during which the basal medium was refreshed every 2–3 days. On day 7, PAFs on basal compartments were co-cultured with PBECs for the next 7 days. On day 14, RNAs from both cell types were collected separately for gene expression assessment.

**Figure 2 cells-11-00526-f002:**
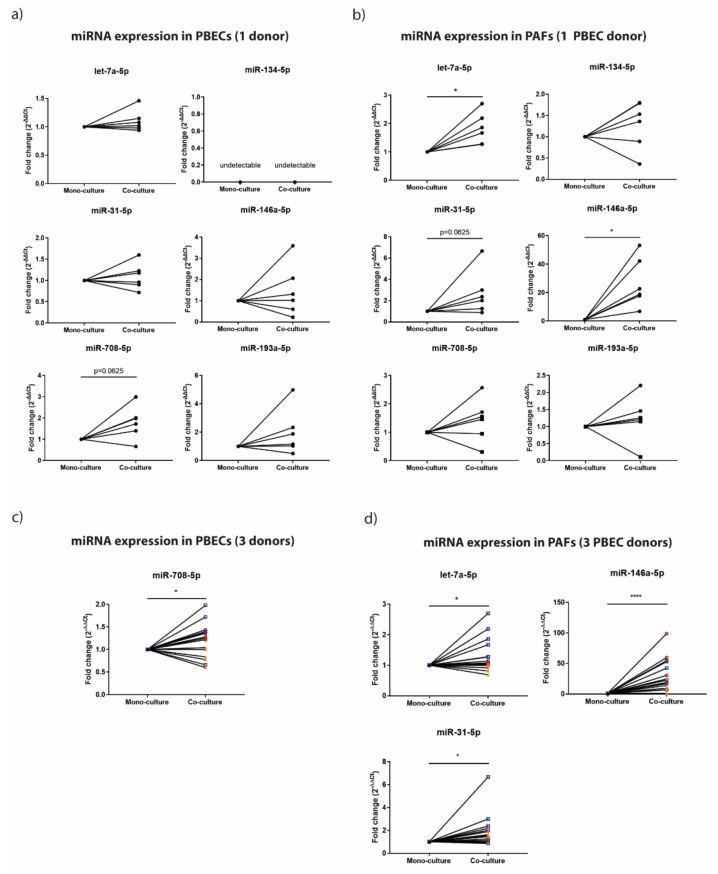
miRNA expression in air–liquid-interface-differentiated primary bronchial epithelial cells (PBECs) and primary airway fibroblasts (PAFs) from severe COPD patients in mono-culture and co-culture. PBECs and PAFs were seeded in duplicates at the apical (epithelial) or basolateral (PAFs) compartment of a transwell system. From day 7 to day 14 upon air exposure of the apical compartment, the cells were cultured in mono-culture or in co-culture. At day 14, total RNA was collected from each cell type separately. (**a**) miRNA expression in PBECs from 1 donor cultured alone and in the presence of PAFs from 6 donors. (**b**) miRNA expression in PAFs from 6 different donors cultured alone and in the presence of PBECs from 1 donor. (**c**) miRNA expression in PBECs from 3 donors cultured alone and in the presence of PAFs from 6 donors. Different symbols indicate different PBEC donors. (**d**) miRNA expression in PAFs from 6 different donors cultured alone and in the presence of PBECs from 3 donors. Different symbols indicate different PBEC donors. Expression of all miRNAs was normalized to the expression of small nuclear RNA, RNU48. Significant differences were determined by the Wilcoxon signed rank test. * *p* < 0.05 and **** *p* < 0.001 between the indicated values.

**Figure 3 cells-11-00526-f003:**
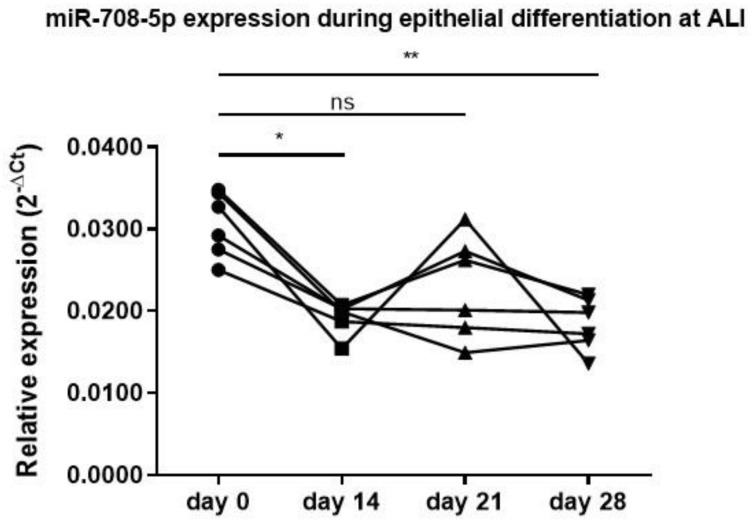
miR-708-5p expression reduces upon mucociliary differentiation. PBECs from 6 non-COPD donors were cultured in air–liquid interface (ALI) for 28 days in the presence of interleukin (IL)-13 (1 ng/mL). Expression of miR-708-5p was normalized to the expression of small nuclear RNA, RNU48. Significant differences in expression in comparison to day 0 were determined by repeated measures 1-way ANOVA with Dunn’s multiple comparison correction. * *p* < 0.05 and ** *p* < 0.01 between the indicated values. ns = not significant.

**Figure 4 cells-11-00526-f004:**
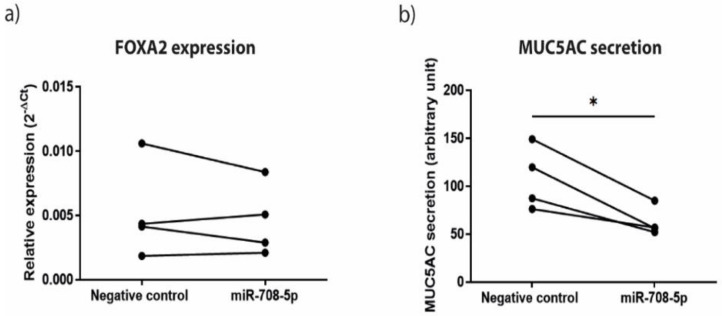
Effect of transfection with miR-708-5p mimic on FOXA2 expression and MUC5AC secretion by A549 epithelial cells. A549 cells were seeded on 24-well plates and transfected with miR-708-5p mimics or negative control siRNA (1 nM) in A549 cells in serum free medium. Cells were harvested for RNA isolation and cell-free supernatants were collected after 24 h. (**a**) Expression of FOXA2 was normalized to the housekeeping genes PPIA and B2M and compared between treatment with oligo control and miR-708-5p mimics. (**b**) MUC5AC levels were measured in cell-free supernatants by ELISA and compared between treatment with oligo control and miR-708-5p mimics. Four experiments were performed independently. Significant differences were determined by paired Student’s *t*-test. * *p* < 0.05 between the indicated values.

**Figure 5 cells-11-00526-f005:**
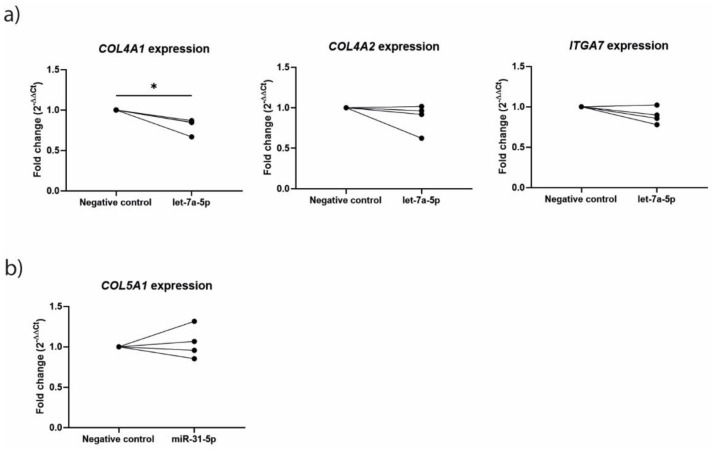
Transfection with let-7a-5p and miR-31-5p mimic in normal human lung fibroblasts (NHLFs). NHLFs were seeded in 24-well plates and transfected with let-7a-5p and miR-31-5p mimics or negative control siRNA (5 nM) and harvested for RNA isolation after 24 h. (**a**) The expression of let-7a-5p’s predicted targets COLA41, COL4A2 and ITG7 was normalized to glyceraldehyde 3-phosphate dehydrogenase (GAPDH), comparing NHLFs treated with let-7a-5p mimic and negative oligo control. (**b**) The expression of miR-31-5p’s predicted target COL5A1 comparing NHLFs treated with miR-31-5p mimic and negative oligo control. Four experiments were performed independently. Significant difference was determined by paired Student’s t-test. * *p* < 0.05 between the indicated values.

**Table 1 cells-11-00526-t001:** Patient characteristics.

Donor	Age	Gender	Pack-Years	FEV1 (%predicted)	FEV1/FVC
E1	58	m	35	15	0.19
E2	49	m	11	20	0.22
E3	60	f	40	18	0.19
F1	53	f	40	23	0.26
F2	48	f	30	12	0.26
F3	59	m	40	15	0.29
F4	61	f	30	16	0.19
F5	58	m	35	15	0.19
F6	59	m	47	15	0.21

All patients were ex-smokers with COPD stage IV with CMH. E1–E3 are PBEC donors; F1–F6 were PAF donors; FEV1 is forced expiratory volume in 1 s; FVC is forced vital capacity; CMH is chronic mucus hypersecretion defined by clinical records; m is male, f is female.

**Table 2 cells-11-00526-t002:** Primer Sequences.

Gene	Forward/Reverse	Sequence
*GAPDH* [14]	Forward	CCCTTCATTGACCTCAACTACA
	Reverse	ATGACAAGCTTCCCGTTCTC
*COL1A1* [15]	Forward	GGAATGAAGGGACACAGAGGTT
	Reverse	AGTAGCACCATCATTTCCACGA
*COL4A1* [15]	Forward	CAGGCACCCCATCTGTTGAT
	Reverse	CATTGCCTTGCACGTAGAGC
*COL4A2* [15]	Forward	TTATGCACTGCCTAAAGAGGAGC
	Reverse	CCCTTAACTCCGTAGAAACCAAG
*COL5A1* [16]	Forward	GCCCGGATGTCGCTTACAG
	Reverse	AAATGCAGACGCAGGGTACAG
*ITGA7* [17]	Forward	GCTGTGAAGTCCCTGGAAGTGATT
	Reverse	GCATCTCGGAGCATCAAGTTCTT

**Table 3 cells-11-00526-t003:** CMH-associated miRNAs identified in bronchial biopsies of COPD patients.

miRNA Positively Associated with CMH	miRNA Negatively Associated with CMH
let-7a-5p	**miR-134-5p**
let-7d-5p	**miR-146a-5p**
let-7f-5p	**miR-193a-5p**
miR-31-5p	miR-500a-3p
miR-708-5p	miR-1207-5p

The miRNAs selected for qPCR are indicated in bold.

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
