# Peer review of "MicroRNAs Associated with Chronic Mucus Hypersecretion in COPD Are Involved in Fibroblast–Epithelium Crosstalk"

_cells, 2022, doi:10.3390/cells11030526_

Round 1

Reviewer 1 Report

I have read with interest the manuscript by Tasena et al. about the role of miRNAs in the CMH in COPD. Overall, the study design is appropriate, laboratory methodology is precisely described and it is obvious, that authors are experts in the field with publication history related to the topic. Study also promotes futher research as potential use of miRNAs as therapy for CMH in COPD may provide novel benefits to the patients, e.g. focusing on miR-707. However, I need some clarification about how the patients data were collected (this may be just my misunderstanding and it does not necessarily imply any severe ethical problem – see comments bellow) and patients description needs to be improved if possible. Furthermore, the use of only 1 sample as „monocultured control“ raises methodological/statistical bias, that shall be more commented. Specific comments to be answered by authors can be found bellow.

Comments:

  1. Line 85: Sentence „BECs were isolated from residual bronchial tissue of 6 non-COPD transplant donors of whom no further information was available“ raises methodological question whether observed differences in miRNA expression are not affected by age, obesity or other conditions that are unknown due to the fact, that there are no characteristics of this cohort. Please, clarify or comment which patients served as donors (Healthy after accident? Smokers? Non-smokers? Opted-in or opted-out patients?) and try to trace more specific characteristics of them.
  2. Table 1: Patient characteristics in the Table are not satisfactory, there are no information about comorbidities, weight, height, BMI – all of which could affect miRNA expression. Please, try to update.
  3. Line 93: Cornerstone of your work is focus on mucous hypersecretion and the only definition of CMH is the sentence „The presence of CMH was based on clinical records, with patients indicating daily cough and production of mucus.“ This is not sufficient. Was this information collected prospectively or retrospectively? Was it collected via standardized questionnaire? Also, this raises ethical concern, whether patient agreed with the use of their medical records according to GDPR. Please, explain.
  4. Line 138: Please, comment (with a reference) why RNU48 was selected as internal control and no other RNU.
  5. For Figure 2c) and 2d) please explain, why there is just one point for 3 PEBCs measurements (I would expect 3 points in the graph, or the use of non-linear point graph for comparison). Please also comment why more monocultures were not used in the whole experiment, as comparing 1 measurement with 6 measurement seems statistically inappropriate as just one monoculture measurement could totally bias the data (e.g. if the one measurement would be extreme, all other results may be flawed).
  6. Please, unify the use of abbreviations, i.e. introduce them on the first occasion and then stick to their use and do not introduce them again. Eg.
    • Line 73: Abbreviation ALI is not explained at first use
    • Line 168: Abbreviations „FOXA2, TMEM88, B2M and PPI“ are not explained at their first use.
    • Line 176: Abbreviation MUC5AC is not explained at first use
    • Line 342: Abbreviation COL4A1 is introduced in discussion however, it has already been repetitively used in the previous parts of the manuscript.
  7. When talking about miRNA mimics transfection, please refer to it as transfection and not miRNA mimic overexpression, eg. Line 164: Please rephrase „To assess mRNA expression in A549 cells overexpressing miRNA mimic“ to „To assess mRNA expression in A549 cells TRANSFECTED WITH miRNA mimic“

Author Response

I have read with interest the manuscript by Tasena et al. about the role of miRNAs in the CMH in COPD. Overall, the study design is appropriate, laboratory methodology is precisely described and it is obvious, that authors are experts in the field with publication history related to the topic. Study also promotes futher research as potential use of miRNAs as therapy for CMH in COPD may provide novel benefits to the patients, e.g. focusing on miR-707. However, I need some clarification about how the patients data were collected (this may be just my misunderstanding and it does not necessarily imply any severe ethical problem – see comments bellow) and patients description needs to be improved if possible. Furthermore, the use of only 1 sample as „monocultured control“ raises methodological/statistical bias, that shall be more commented. Specific comments to be answered by authors can be found bellow.

Response: We thank the reviewer for the careful evaluation of our manuscript, considering it well-conducted, and for the useful suggestions for improvement. We have addressed the specific comments below. With respect to the mono-cultured control, we would like to stress that each fibroblast donor (n=6) and each epithelial donor (n=3) had its own mono-cultured control. Because we expressed the mRNA data as fold induction (using the 2^-deltadeltaCt calculation) all control values were set at 1.

 Major Comments:

  • Line 85: Sentence „BECs were isolated from residual bronchial tissue of 6 non-COPD transplant donors of whom no further information was available“ raises methodological question whether observed differences in miRNA expression are not affected by age, obesity or other conditions that are unknown due to the fact, that there are no characteristics of this cohort. Please, clarify or comment which patients served as donors (Healthy after accident? Smokers? Non-smokers? Opted-in or opted-out patients?) and try to trace more specific characteristics of them.

 Response: The epithelial cells were isolated from residual bronchial tissue of lungs that were used for lung transplantation, i.e. from the transplant donors. This means that the lungs were in good condition and approved for transplantation. However we cannot exclude that donors have never smoked during their life. For privacy reasons, we are not able to obtain any information on these subjects. Therefore, we refer to these donors as non-COPD controls. We agree with the reviewer that age and other personal factors can influence the expression of miRNAs and lead to differences in their tissue expression between individuals. This may explain differences in expression of miR-708 observed at baseline in Figure 3. However, our aim here was not to compare different groups (e.g. COPD to non-COPD) but to assess effects of epithelial differentiation.

  • Table 1: Patient characteristics in the Table are not satisfactory, there are no information about comorbidities, weight, height, BMI – all of which could affect miRNA expression. Please, try to update.

Response: We have provided the most relevant information, i.e. age, gender, smoking status and lung function in table 1. Information on comorbidities, weight, height or BMI is not available for each patient. We agree with the reviewer that age and other personal factors can influence the expression of miRNAs and lead to differences in their tissue expression between individuals, however, our aim in this study was not to compare different groups (e.g. COPD to non-COPD) but to assess effects of fibroblast-epithelial co-culture.

  • Line 93: Cornerstone of your work is focus on mucous hypersecretion and the only definition of CMH is the sentence „The presence of CMH was based on clinical records, with patients indicating daily cough and production of mucus.“ This is not sufficient. Was this information collected prospectively or retrospectively? Was it collected via standardized questionnaire? Also, this raises ethical concern, whether patient agreed with the use of their medical records according to GDPR. Please, explain.

 Response:  We understand the reviewer’s concern and acknowledge that the definition of chronic mucus hypersecretion (CMH) is notoriously difficult. While there are various ways to define CMH, no single definition is perfect. For instance, a quantification of goblet cell numbers (e.g. PAS-scores) may be considered more objective than using questionnaires, but the PAS-scores only reflect goblet cells in a highly specific location of the airways and do not necessarily represent the overall mucus secretion. This inconsistency between the PAS-scores and the patients’ responses to the questionnaires was also shown in our COPD cohort (Tasena et al. Eur Respir J 2018; 52: 1701556). Nevertheless, many symptom-based CMH studies have been performed and delivered positive results in the past (e.g. Allinson, et al. Am J Respir Crit Care Med. 2016;193:662–72, Dijkstra, et al. PLoS One. 2014;9:e91621, and Vestbo et al. Am J Respir Crit Care Med. 1996;153:1530–1535). In this study, the information on CMH was collected retrospectively from the clinical records of the patients, which were checked for information regarding complaints and symptoms on daily cough and sputum production. For this study we only selected patients of whom information on CMH symptoms was available. We have now added this to the Methods (lines 96-99). As the cells were derived from left-over lung material after lung surgery and transplant procedures, no specific ethical approval and informed consent was obtained and therefore also no questionnaire data on CMH was available.

  • Line 138: Please, comment (with a reference) why RNU48 was selected as internal control and no other RNU.

Response: We have based our selection of RNUs on previous findings, selecting the RNU that was most stable in the co-cultures. We have initially assessed the expression of RNU24, RNU46, RNU48 and RNU49, which are internal controls commonly used for miRNA studies, in previous pilot co-culture experiments. We found that RNU46 and RNU49 were barely expressed in both epithelial cells and fibroblasts. While RNU24 and RNU48 were both expressed stably across multiple samples, the expression level of RNU48 was slightly higher than RNU24. Therefore, we have chosen RNU48 as internal control throughout the experiments. We have now referred to previous publications  from our group where we used this RNU (Tasena et al. Eur Respir J 2018; 52, 1701556; Osei et al 2017; 49, 1602538).

  • For Figure 2c) and 2d) please explain, why there is just one point for 3 PEBCs measurements (I would expect 3 points in the graph, or the use of non-linear point graph for comparison). Please also comment why more monocultures were not used in the whole experiment, as comparing 1 measurement with 6 measurement seems statistically inappropriate as just one monoculture measurement could totally bias the data (e.g. if the one measurement would be extreme, all other results may be flawed).

Response: We would like to stress that each fibroblast donor (n=6) and each epithelial donor (n=3) had its own mono-cultured control. Because we expressed the mRNA data as fold induction (using the 2^-deltadeltaCt calculation) all control values are set at 1. Our experimental set-up was designed to address the hypothesis that fibroblasts regulate mucus hypersecretion and that miRNAs are involved in fibroblast-epithelial crosstalk. Therefore, we used 6 different fibroblast donors and kept the epithelial component stable. In order to exclude the possibility of a one-donor bias, we included two additional epithelial donors, which were each cultured with 6 different fibroblast donors. We have added the data as 2^-deltaCt for the reviewer’s convenience in figure 1 of this letter. This shows that there is donor variation in the baseline expression of some of the miRNAS (especially epithelial miR-708-5p) and/or in the response to co-culture (especially for let-7a-5p expression), but overall we observed consistent effects. We feel that showing the data as fold induction compared to the monoculture and combining them in 1 graph conveys our message most clearly. 

  • Please, unify the use of abbreviations, i.e. introduce them on the first occasion and then stick to their use and do not introduce them again. Eg. Line 73: Abbreviation ALI is not explained at first use Line 168: Abbreviations „FOXA2, TMEM88, B2M and PPI“ are not explained at their first use. Line 176: Abbreviation MUC5AC is not explained at first use Line 342: Abbreviation COL4A1 is introduced in discussion however, it has already been repetitively used in the previous parts of the manuscript.

Response: We apologize for this error and have now explained the ALI abbreviation (air-liquid interface) at first use (line 75). The other names refer to gene names, but nevertheless, we have now written them in full the first time for all genes and for MUC5A.

  • When talking about miRNA mimics transfection, please refer to it as transfection and not miRNA mimic overexpression, eg. Line 164: Please rephrase „To assess mRNA expression in A549 cells overexpressing miRNA mimic“ to „To assess mRNA expression in A549 cells TRANSFECTED WITH miRNA mimic“

Response: We thank the reviewer for the useful suggestion and have now adapted this throughout the Methods and Results sections.

Reviewer 2 Report

In this article Tasena and colleagues examine the role of growing bronchial epithelial cells in co-culture with fibroblasts from patients with COPD with a chronic mucus hypersecretion phenotype. They show that in co-culture miRNAs are differentially expressed, showing that miR-708-5p, a miRNA known to be elevated in bronchial biopsies from CMH patients, is also elevated in PBEC when grown in co-culture. This miRNA however is decreased during ALI differentiation when a CMH phenotype is induced in healthy cells and may also negatively regulate mucus release. Although of interest, it is difficult in its current form to understand how the co-culture drives the changes in the expression of these miRNAs and how these miRNAs are directly having a function on this phenotype. Also it is not clear that growing these cells together further drives a CMH phenotype and this occurs via cross talk between these cells via these miRNAs.

Major:

  1. Was 14 days of differentiation enough to full differentiate your cells at ALI? Usual protocols are from 21-28 days (as used for your healthy PBECs) and it is known that COPD epithelial cells differentiate more slowly than those from healthy individuals. How did you define full differentiation of these cells?

  1. What media were the cells grown in at co-culture? This is important as epithelial cell media has a tendency to inhibit/kill fibroblasts and if fibroblast media was used, how did this effect differentiation at ALI?

  1. Currently, in this work there is no evidence that in your co-culture system there is an increase in any markers of CMH. This should be the first experiment performed as the hypothesis is that co-culture drives miRNA expression which drives mucus secretion/ increased mucus producing cells. Does growing the cells together increase mucus gene expression in the PBECs or increase MUC5AC release into the media when these cells are grown together?

  1. There is no mechanism as to how co-culture may be inducing the expression of these miRNAs and this is not discussed within the paper. Do the authors think inflammatory factors release by a cell may induce/supress these miRNAs or are they transported to one another in vesicles?

  1. At baseline without co-culture are the miRNAs more expressed in one cell type compared to another? As you have non-CMH and CMH fibroblasts and epithelial cells are any of the miRNAs altered at baseline between the disease phenotypes? Do the individual cells replicate what is seen in the bronchial brushings?

  1. Did the authors examine whether growing COPD PAFs with non-CMH PBECs drove any changes in the expression of your selected miRNAs or COPD PBECs with non-CMH PAFs? If these miRNAs are already elevated in these cells it maybe that they cannot be induced further?

  1. In figure 2 D, is let-7a-5p increase only occurring when PAFs are co-cultured with one PBEC and this appears to be the one displayed in figure 2B? Does this suggest a lack of reproducibility in the induction as this only occurred in 1 of the 3 PBEC donors, as the other 2 donors have little effect on let-7a-5p expression?

  1. Figure 3. This figure is difficult to interpret without the use of a control set of ALI samples which are not treated with IL-13. Also, if miR-708-5p is elevated in CMH patients and IL-13 is used to drive this phenotype, then why would miR-708-5p decrease during differentiation as you would expect the opposite? Or do these findings suggest that the increase in miR-708-5p in not sufficient to have an effect on mucus secretion in patients with CMH? Finally, you are examining multiple time points across an experiment and therefore performing Wilcoxon signed rank test is not an appropriate statistical test. There are multiple comparisons on your graph and therefore this should be considered.

  1. Does IL-13 induce or decrease miR-708-5p expression?

  1. Although as stated in the paper, technically difficult, to really understand the role of miR-708-5p in CMH, this miRNA needs to be over-expression during differentiation to fully understand whether it is having a positive effect or negative effect on this phenotype.

  1. Figure 5. Why were the certain collagen genes selected for each miRNA? Was any of the targets of these miRNA examined and this the reason? Was prediction software used? Have any of the targets selected been shown to be direct targets of any of these miRNAs or are these down-stream effects of the miRNAs rather than direct targets?

Minor:

  1. Why was a siRNA scramble control used rather than a miRNA mimic scramble? These will undergo different processing within the cell making it not a true control for the transfection

  1. In table 3, only miR-134-5p, miR-146a-5p and miR-193a-5p are highlighted in bold to represent which miRNAs were selected, but you also examine Let-7a-5p, miR-31-5p and miR-708-5p. These should be highlighted in bold.

Author Response

In this article Tasena and colleagues examine the role of growing bronchial epithelial cells in co-culture with fibroblasts from patients with COPD with a chronic mucus hypersecretion phenotype. They show that in co-culture miRNAs are differentially expressed, showing that miR-708-5p, a miRNA known to be elevated in bronchial biopsies from CMH patients, is also elevated in PBEC when grown in co-culture. This miRNA however is decreased during ALI differentiation when a CMH phenotype is induced in healthy cells and may also negatively regulate mucus release. Although of interest, it is difficult in its current form to understand how the co-culture drives the changes in the expression of these miRNAs and how these miRNAs are directly having a function on this phenotype. Also it is not clear that growing these cells together further drives a CMH phenotype and this occurs via cross talk between these cells via these miRNAs.

Response: We thank the reviewer for the careful evaluation of our manuscript, considering or data of interest,  and for the useful suggestions for improvement. We have addressed the specific comments below.

Major comments:

  • Was 14 days of differentiation enough to full differentiate your cells at ALI? Usual protocols are from 21-28 days (as used for your healthy PBECs) and it is known that COPD epithelial cells differentiate more slowly than those from healthy individuals. How did you define full differentiation of these cells?

Response: We understand the concern of the reviewer, but as mentioned on line 108, we used a previously published protocol (Spanjer et al, Thorax 2016; 71, 312–322; reference 10 of our manuscript), where cells were already treated with retinoic acid before air exposure, leading to differentiation towards mucus-producing and ciliated cells within 14 days. This was previously confirmed by secretion of mucins and expression of ciliated cell markers. In our experiments, successful differentiation was confirmed by the expression of mucus and substantial increase in trans-epithelial resistance (data not show). We have now mentioned this in the Methods section on lines 108-109.

  • What media were the cells grown in at co-culture? This is important as epithelial cell media has a tendency to inhibit/kill fibroblasts and if fibroblast media was used, how did this effect differentiation at ALI?

Response: As mentioned on line 122 in the Methods, we used ALI medium (DMEM and BEBM in 1:1 ratio supplemented with a set of BEGM Single Quots, 1.5µg/ml BSA (Sigma-Aldrich) and 15 ng/ml retinoic acid) during the co-cultures. We performed several optimization experiments to determine the optimal cell culture conditions, where we also assessed viability of both cell types. With the optimal culture conditions, we were able to culture the cells together for at least one week without affecting the viability of any of the cell types.

  • Currently, in this work there is no evidence that in your co-culture system there is an increase in any markers of CMH. This should be the first experiment performed as the hypothesis is that co-culture drives miRNA expression which drives mucus secretion/increased mucus producing cells. Does growing the cells together increase mucus gene expression in the PBECs or increase MUC5AC release into the media when these cells are grown together?

Response: This is indeed a relevant question and the subject of another manuscript that has recently been submitted for consideration of publication in a peer-reviewed journal. As mentioned in our Introduction, our group previously observed that fibroblast mediators increase mucus production by epithelial cells upon 24 hours of co-culture (Spanjer et al, Thorax 2016; 71, 312–322) and in our current model with prolonged co-culture we also observed an increase in MUC5AC (data yet unpublished). 

  • There is no mechanism as to how co-culture may be inducing the expression of these miRNAs and this is not discussed within the paper. Do the authors think inflammatory factors release by a cell may induce/supress these miRNAs or are they transported to one another in vesicles?

Response: We agree that this is an interesting question, although beyond the scope of the current manuscript.  We have previously shown that epithelial release of IL-1alpha induces the upregulation of miR-146a-5p in fibroblasts upon their co-culture, as described in the Introduction (lines 63-64). Whether a similar mechanism may be involved in the regulation of the other CMH-associated miRNAs that were differently expressed upon co-culture needs further investigation. Transfer of miRNAs to another cell type via extracellular vesicles is also a well-known mechanism of intercellular communication. We have now mentioned this in the Discussion section, lines 415-421.

  • At baseline without co-culture are the miRNAs more expressed in one cell type compared to another? As you have non-CMH and CMH fibroblasts and epithelial cells are any of the miRNAs altered at baseline between the disease phenotypes? Do the individual cells replicate what is seen in the bronchial brushings?

Response: These data have been published in our previous manuscript as referred to in our Results section on line 239 (reference 2). Let-7a-5p was considerably higher expressed in fibroblasts. Epithelial cells did not express miR-134-5p. Other miRNAs were expressed in each cell type. We only found a significant difference between non-CMH and CMH fibroblasts for miR-134-5p, being lower expressed with CMH.

  • Did the authors examine whether growing COPD PAFs with non-CMH PBECs drove any changes in the expression of your selected miRNAs or COPD PBECs with non-CMH PAFs? If these miRNAs are already elevated in these cells it maybe that they cannot be induced further?

Response: We agree that this is an interesting question. However, assessing whether co-culture with PBECs  from controls or COPD patients with or without CMH differently affects miRNA expression in fibroblasts from controls or COPD patients with/or without COPD would result in ALI cultures from 144 different combinations instead of the 18 different combinations that we currently describe. For this manuscript, we decided to first focus on the role of CMH-associated miRNAs in epithelial-fibroblast crosstalk, future studies are needed to investigate the additional effect of disease and CMH state. As mentioned above, our previous study has demonstrated that only miR-134-5p was found to be differently expressed at baseline, with lower expression in fibroblasts from COPD patients with compared to those without CMH.

  • In figure 2 D, is let-7a-5p increase only occurring when PAFs are co-cultured with one PBEC and this appears to be the one displayed in figure 2B? Does this suggest a lack of reproducibility in the induction as this only occurred in 1 of the 3 PBEC donors, as the other 2 donors have little effect on let-7a-5p expression?

Response: Indeed we observed the most robust increase upon co-culture with the donor shown in figure 2B. After validation of these findings using 2 additional epithelial donors, we also found an increase in the donor depicted by the red dots, whereas the donor depicted in the orange triangles did not show an increase, which can also be appreciated in figure 1 of this letter. However, in the combined statistical analysis with all three donors the effect remained significant. 

  • Figure 3. This figure is difficult to interpret without the use of a control set of ALI samples which are not treated with IL-13. Also, if miR-708-5p is elevated in CMH patients and IL-13 is used to drive this phenotype, then why would miR-708-5p decrease during differentiation as you would expect the opposite? Or do these findings suggest that the increase in miR-708-5p in not sufficient to have an effect on mucus secretion in patients with CMH? Finally, you are examining multiple time points across an experiment and therefore performing Wilcoxon signed rank test is not an appropriate statistical test. There are multiple comparisons on your graph and therefore this should be considered.

 Response: Similar to EGF (one of the BEGM Single Quots factors), IL-13 is often added during ALI culture to support goblet cell differentiation. Unfortunately, we have not performed ALI cultures where either one or both factors were left out. As explained in our Discussion section (lines 376-379), we indeed previously found that miR-708-5p was positively associated with CMH, and we therefore speculate that miR-708 upregulation in CMH may act as a compensatory mechanism in the interaction between epithelial cells and fibroblasts as an attempt to suppress mucus hypersecretion, a mechanisms that may fail in CMH. However, whether and why this is the case needs further investigation. As for the statistical testing, we agree that 1-way ANOVA with correction for multiple testing is a more appropriate test, which we have now performed (See new Figure 3 and the description in the legend in our manuscript). This did not change the overall message, although the effect was no longer significant at day 21, yet more significant at day 28.

  • Does IL-13 induce or decrease miR-708-5p expression?

Response: To the best of our knowledge, there are no data to show the effect of IL-13 on miR-708-5p expression.

  • Although as stated in the paper, technically difficult, to really understand the role of miR-708-5p in CMH, this miRNA needs to be over-expression during differentiation to fully understand whether it is having a positive effect or negative effect on this phenotype.

Response: This is a useful suggestion. However, in 5 different attempts, we were not able to substantially overexpress the mimic in ALI-cultured cells. We have now included the data on the expression of miR-708 upon transfection with miR-708-5p mimic in ALI cultures (online data supplementary figure 1). We agree that it will be of relevance to know whether miR-708-5p regulates mucus expression and/or differentiation of goblet cells during the full period of ALI culture. However, transfection with mimics is transient by nature and therefore this is not feasible in the current set-up and within the time frame.

  • Figure 5. Why were the certain collagen genes selected for each miRNA? Was any of the targets of these miRNA examined and this the reason? Was prediction software used? Have any of the targets selected been shown to be direct targets of any of these miRNAs or are these down-stream effects of the miRNAs rather than direct targets?

Response: As described in our Results section (lines 333-335) these are known predicted and correlated target genes that were also negatively associated with CMH as demonstrated previously by us (reference 2 of our manuscript).

Minor comments:

  • Why was a siRNA scramble control used rather than a miRNA mimic scramble? These will undergo different processing within the cell making it not a true control for the transfection

Response: We thank the reviewer for the useful suggestion. However, the siRNA scrambled control is recommended by Qiagen as negative control for miRNA mimic experiments, see AllStars Negative Control siRNA (qiagen.com). The sequence of this control, and also other commercial miRNA scramble controls, is proprietary and thus unknown to us, so we rely on the manufacturer’s advice on their applications. The scrambled control has been validated extensively, causing minimal to no changes on gene expression and cellular phenotype. It has been used in various published studies (e.g. Philippe et al, J Immunol 2012; 188, 454-461; Qian et al,  Cell Death Discovery 2017; 7028; Alderman et al, Tumor Biology 2016; 37, 13941–13950), and we are confident of its reliability. We have used this control also for all our previously published mimic experiments with good consistency. 

  • In table 3, only miR-134-5p, miR-146a-5p and miR-193a-5p are highlighted in bold to represent which miRNAs were selected, but you also examine Let-7a-5p, miR-31-5p and miR-708-5p. These should be highlighted in bold.

Response: We fully agree, and hence these were already highlighted in bold. We will ensure that this is clearly visible in the pdf file of our manuscript.

Round 2

Reviewer 1 Report

I have read with interest the revised paper by Tasena et al. Overall the manuscript was greatly improved and I have no other comments considering the content of the manuscript - I sincerely congratulate the authors on conducting such an interesting study. However, one of my commnets was not adequatelly addressed and needs further explanation, as this involves the publication /research conductance ethics.

In your response to my comment about CMH definition and ethics you state:

"We understand the reviewer’s concern and acknowledge that the definition of chronic mucus hypersecretion (CMH) is notoriously difficult. While there are various ways to define CMH, no single definition is perfect. For instance, a quantification of goblet cell numbers (e.g. PAS-scores) may be considered more objective than using questionnaires, but the PAS-scores only reflect goblet cells in a highly specific location of the airways and do not necessarily represent the overall mucus secretion. This inconsistency between the PAS-scores and the patients’ responses to the questionnaires was also shown in our COPD cohort (Tasena et al. Eur Respir J 2018; 52: 1701556). Nevertheless, many symptom-based CMH studies have been performed and delivered positive results in the past (e.g. Allinson, et al. Am J Respir Crit Care Med. 2016;193:662–72, Dijkstra, et al. PLoS One. 2014;9:e91621, and Vestbo et al. Am J Respir Crit Care Med. 1996;153:1530–1535). In this study, the information on CMH was collected retrospectively from the clinical records of the patients, which were checked for information regarding complaints and symptoms on daily cough and sputum production. For this study we only selected patients of whom information on CMH symptoms was available. We have now added this to the Methods (lines 96-99). As the cells were derived from left-over lung material after lung surgery and transplant procedures, no specific ethical approval and informed consent was obtained and therefore also no questionnaire data on CMH was available."

I thank you for such a good explanation and definition of CMH! But I still have a problem with the need of the Ethical approval. I totally understand that the material used for the study was leftovers of the lungs and thus, there is no need to have ethical approval to use it for the research.

But as you state in you response (and also in the revised version of the manuscript) and as I underlined above, you were using clinical records of the patients. And for any use of the medical record of the patient, there needs to be some approval, not just the informed consent - in our hospital, all patients admitted to the standard wards or visiting out-patient clinics signs that their records can be used anonymously for the research purposes. Is some kind of document also signed in your hospital? If yes, consider my comment answered and wish you bet of luck with your future research!

Author Response

Response:  We thank the reviewer again for the careful evaluation of our manuscript and raising this important issue. The reviewer is correct in that the patient’s approval for use of their anonymised patient record data at present is the standard, also in the Netherlands, according to national law as well as the General Data Protection Regulation (EU) 2016/679 (GDPR). For many patients included, that is part of their informed consent. In our hospital there is ample information about the use of their data to which they can object (opt-out), which is registered, but there is as yet no general procedure for patients to give written consent for use of their anonymized data. For many patients, their used (anonymized) tissues and data were obtained many years ago, and because of their disease, in the majority of cases the patients are not alive anymore. Because of this, these are exempt from approval according to GDPR art. 14, paragraph  5, as the provision of such information proves impossible or would involve a disproportionate effort. Obviously, with this exemption for approval of use of the data, we carefully followed the relevant conditions of GDPR article 9. We have now added to our manuscript that the data have been collected according to the national law as well as the General Data Protection Regulation (EU) 2016/679 (GDPR), lines 93-94.

Reviewer 2 Report

I thank the authors for clearing up the points I had made. 

Author Response

We thank the reviewer again for the careful evaluation of our manuscript and are pleased to see that the reviewer has no further concerns.